# Disrupting Defenses: Effects of Bisphenol A and Its Analogs on Human Antibody Production In Vitro

**DOI:** 10.3390/life15081203

**Published:** 2025-07-28

**Authors:** Francesca Carlotta Passoni, Martina Iulini, Valentina Galbiati, Marina Marinovich, Emanuela Corsini

**Affiliations:** Laboratory of Toxicology and Risk Assessment, Department of Pharmacological and Biomolecular Sciences “Rodolfo Paoletti”, Università degli Studi di Milano, Via Balzaretti 9, 20133 Milan, Italy; francesca.passoni@unimi.it (F.C.P.); valentina.galbiati@unimi.it (V.G.); marina.marinovich@unimi.it (M.M.); emanuela.corsini@unimi.it (E.C.)

**Keywords:** bisphenols, immunoglobulins, PBMCs, in vitro, new approach methodologies (NAMs)

## Abstract

Bisphenol A (BPA) is an endocrine-disrupting chemical with estrogen-like activity, known to impair immune function. BPA may act as a pro-inflammatory agent, reducing immune response efficacy, increasing bacterial load in E. coli infections, and altering immune responses in parasitic infections (Leishmania major, Nippostrongylus brasiliensis, Toxocara canis) through cytokine and regulatory T-cell modulation. Following its ban in food contact materials in Europe, several analogs have been introduced. This study assessed the immunotoxicity of BPA and six analogs, namely BPAP, BPE, BPP, BPS-MAE, BPZ, and TCBPA, by evaluating in vitro the antibody production. Peripheral blood mononuclear cells from healthy male and female donors were exposed to increasing concentrations of each compound for 24 h. After stimulation with rhIL-2 and ODN2006, IgM and IgG secretion were measured on day six. All compounds suppressed antibody production in a concentration-dependent manner, with some sex-related differences. IC_50_ values showed BPP as the most potent suppressor, and BPE as the weakest. Similarly, IC_20_ values confirmed these differences in potency, except for BPA being the weakest for IgM in males. Overall, te results do not support the idea that BPA analogs are safer than BPA.

## 1. Introduction

Endocrine disruptors (EDs) represent a major human and environmental health threat [1]. One well-known ED is bisphenol A (BPA), which is used in the production of polymeric materials, plastic, and epoxy resins to confer strength and resistance to products [2,3,4]. BPA can be found in many consumer products, including toys, personal care products, and thermal paper [5]. Exposure primarily occurs through the ingestion of contaminated food and water. Other exposure routes include inhaling volatilized BPA from household goods, thermal paper, and dental materials, and dermal contact [6]. Studies on the pharmacokinetics of BPA following oral administration suggest that the chemical is absorbed by the gastrointestinal tract and that BPA glucuronide is formed as the main metabolite. Thayer et al. [7] demonstrated that BPA is completely converted into this metabolite after oral administration and is excreted in urine within 24 h. The percentage of unconjugated BPA detected is less than 1%. A second metabolite, BPA-sulfate, has also been detected in urine, accounting for approximately 10% [8]. BPA and its conjugated forms are known to bind to the estrogen receptor (ER) at concentrations as low as 10^−7^ to 10^−15^ M [9]. It also interacts with other nuclear receptors, including the androgen receptor (AR) and the thyroid hormone receptor (THR), affecting the reproductive and immune systems [6]. Quantification studies have reported BPA and some analogs in human plasma and urine samples, with concentrations varying as follows: BPA (0.79–7.12 ng/mL in plasma [10]; 0–4.38 ng/mL in urine [11]), bisphenol AP (BPAP) (0.051 ng/mL [12]; 0.352 ng/mL [13]), bisphenol P (BPP) (0.58 ng/mL [12]; 0.668 ng/mL [13]), and bisphenol S 4-allyl ether (BPS-MAE) (0.0035–0.017 ng/mL in urine [14]). For bisphenol E (BPE), bisphenol Z (BPZ), and 3,3′,5,5′-Tetrachlorobisphenol A (TCBPA), available data are limited or not determined in blood or urine samples.

In 2023, the European Food Safety Authority (EFSA) re-evaluated the tolerable daily intake of BPA, reducing it from 4 μg/kg/body weight (bw) per day to 0.2 ng/kg/bw. This decision was based on observational studies that identified the immune system as being particularly sensitive to BPA exposure. The critical endpoint that led to this revision was an increase in T helper (Th) 17 cells and interleukin (IL) 17, both of which are associated with inflammation in autoimmune diseases [15]. In addition, studies on prenatal BPA exposure reveal that it can increase the risk of childhood asthma and wheezing [16,17], alter metabolic parameters, and increase the risk of metabolic diseases such as obesity [18]. Given that endocrine hormones play a central role in immune regulation, EDs such as BPA may alter immune outcomes via various molecular and physiological mechanisms [19]. Several studies indicate that the physiological functions of immune system cells, such as the production of tumor necrosis factor-α (TNF-α) and IL-6 in macrophages, can be disrupted by BPA exposure at concentrations of 0.01 and 0.1 µM. BPA can also impair the differentiation and maturation of human monocytes into monocyte-derived dendritic cells (MDDCs) in vitro at a dose of 50 µM [20,21,22]. Regarding the effects of BPA on B cells, most studies have been conducted in vivo, while in vitro research remains scarce. Both animal and human studies indicate that BPA and its analogs can modify immunoglobulin (Ig) release relative to normal conditions. In animal in vivo studies, Yanagisawa et al. showed increased levels of IgE and IgG following oral administration (0.4 µg/kg bw per day) of Bisphenol S (BPS) [23] and of BPA (0.09 µg/kg bw per day) [24] in BALB/c mice. Huang et al. [25] demonstrated that lower doses (250 µg/kg bw) of BPA induced low reactivity IgM production in mice [26]. Moreover, in human observational studies, BPA exposure (detected as BPA glucuronide in urine at levels of 11.84 ng/mL and 8.84 ng/mL at ages 3 and 6 years, respectively) has been associated with increased IgE levels during childhood and may contribute to the pathogenesis of allergic asthma, with girls appearing more susceptible than boys [27]. Furthermore, a high dose of BPA (100 µM) was found to decrease B-cell viability by approximately 70% in vitro human B lymphoblast cells [28]. In summary, the available evidence collectively suggests that BPA can increase or suppress Ig levels, depending on the dose, developmental timing, and species studied. In animals, BPA has been linked to elevated levels of IgG and IgE, whereas human studies have associated higher levels of urinary BPA with altered levels of IgE during childhood. Together, these findings support the hypothesis that exposure to BPA may lead to immunoglobulin dysregulation, thereby contributing to the development of allergic or immune-related pathologies.

Immune modulation induced by BPA may impair the body’s defense mechanisms and antibody responses to vaccination. Various studies on different types of infection have shown that BPA can influence host immunity by modulating immune cells’ function, but it can also impair the immune response and facilitate pathogen persistence. For example, research by Sugita-Konishi et al. [29] demonstrates that exposure to BPA impairs the ability of young mice to eliminate Escherichia coli, inhibits neutrophil phagocytosis, and reduces monocyte and lymphocyte populations. In addition to studies on bacterial infections, the effects of BPA on parasitic infections have also been investigated. Prenatal oral administration of BPA to mice infected with Leishmania major causes foot pad inflammation and a decrease in the number of T-regulatory cells [30], as well as increasing susceptibility to Toxocara canis infection [31]. In contrast, the effects on Trichinella spiralis seem protective, with a decrease in the number of larvae observed [32]. Regarding the impact on the efficacy of vaccinations, Uhm and Kim’s [33] research focuses on the development of the immune response to the hepatitis B virus (HBV). BPA exposure may lead to increased susceptibility to HBV, even after vaccination, demonstrating that BPA affects the proper functioning of the immune system.

In recent years, restrictions on the use of BPA have led to the synthesis of new bisphenol analogs, which are currently less regulated. The molecular structure of BPA is characterized by a tetrahedral carbon atom bound to two phenol groups and two methyl groups (C_15_H_16_O_2_, molecular weight 228.29 g/mol) [34]. Analogs share a similar chemical structure and physicochemical properties to BPA, raising concerns about their potential ED properties [35,36]. Due to the restrictions and the lack of knowledge regarding the effects of BPA analogs, these substances have been classified as priority chemicals. Within the framework of the Partnership for the Assessment of Risks from Chemicals (PARC) project, research has focused on studying their impact on the immune system. One of the least explored parameters is their effect on Ig release. To address this knowledge gap, we have used an in vitro human model to investigate antibody secretion, specifically IgG and IgM [37,38,39]. To approximate human exposure more accurately, we selected a broad concentration range, from the highest non-cytotoxic dose to the lowest, which falls within the ng/mL range observed in human plasma and urine. This lower concentration aligns with values reported in biomonitoring studies, thereby enhancing the physiological relevance of the in vitro exposures. Starting from the concentrations corresponding to the 80% of cell viability (CV80s), we performed a 1:10 dilution, resulting in a range of concentrations from µM to nM, corresponding to real-life exposure conditions [40,41]. As an experimental model, peripheral blood mononuclear cells (PBMCs) obtained from healthy donors were used to investigate the effect of BPA and six BPA analogs on Ig secretion. Our work aimed to address the lack of in vitro tools to study the immunotoxicity of BPA and its analogs, as highlighted in the review by Mhaouty-Kodja et al. [42]. The effects were investigated in PBMCs obtained from both male and female donors in order to evaluate also possible sex-specific effects. Our focus was on antibody production by B cells, as they are a fundamental component of the adaptive immune response. Dysregulation of this process can lead to pathological outcomes, such as the production of autoantibodies associated with autoimmune diseases, elevated IgE levels linked to allergic conditions, or, conversely, impaired antibody production, which increases susceptibility to infections.

## 2. Materials and Methods

### 2.1. Chemicals

The BPA analogs selected as priority in the frame of the PARC project [43] are listed in Table 1. In Table 1, CAS numbers, chemical structure, molecular weights, and LogP are also reported.

BPA and its analogs were purchased from Chiron AS (Trondheim, Norway) at the highest available purity, while dimethyl sulfoxide (DMSO; CAS # 67-68-5, ≥ 99.5% purity) and rapamycin (selected as positive control, CAS # 53123-88-9) were purchased from Sigma-Aldrich (St. Louis, MO, USA). All chemicals were diluted in DMSO to make a stock solution, which was stored at −20 °C, whereas the working concentrations were freshly prepared by diluting the stock solution for each treatment. The final concentration of DMSO in the cell culture was 0.1%, which was used as a vehicle control. Recombinant human interleukin 2 (rhIL-2) and the Class B CpG oligonucleotide (ODN) 2006 were purchased from Miltenyi Biotech (Bergisch Gladbach, Germany) and were dissolved in Dulbecco’s phosphate-buffered saline (PBS, Sigma-Aldrich).

### 2.2. Cells

PBMCs were obtained from anonymous buffy coat healthy donors of both sexes, purchased from Niguarda Hospital (Milan, Italy). Following Ficoll gradient centrifugation, PBMCs were washed with PBS and resuspended in RPMI-1640 without phenol red supplemented with 2 mM L-glutamine, 100 IU/mL penicillin, 0.1 mg/mL streptomycin, 10 µg/mL gentamicin, 50 µM 2-mercaptoethanol, and 5% of heat-inactivated dialyzed fetal bovine serum (d-FBS) (complete medium). The culture medium and the supplements were all purchased from Sigma-Aldrich.

### 2.3. Determination of the Range of Concentrations

To identify a non-cytotoxic concentration range to be used (i.e., to select the CV80), the cells were treated with increasing concentrations of the selected compound for 24 h, starting from the highest possible solubility. After this exposure period, the cells were collected, centrifuged for five minutes at 1200 rpm, and resuspended in PBS containing propidium iodide (PI, 1:1000 dilution) (CAS #25535-16-5, Sigma-Aldrich). The samples were then analyzed using a NovoCyte 3000 flow cytometer, and the data were processed using NovoExpress 1.6.1 software (ACEA Biosciences, Inc., San Diego, CA, USA).

### 2.4. Cell Treatment

To evaluate Ig production, PBMCs (1.26 × 10^6^ cells/mL) were plated in 48-well plates with complete medium and exposed to increased concentrations of the selected chemicals and to rapamycin 2 ng/mL and incubated at 37 °C in 5% CO_2_ for 24 h. The cells were then stimulated or not with 20 ng/mL rhIL-2 and 1 µg/mL ODN2006 and incubated at 37 °C in 5% CO_2_ for an additional 6 days as described by Tuijnenburg et al. (2020) [37].

### 2.5. Cell Viability Analysis

To assess possible cytotoxicity, the CyQUANT™ LDH Cytotoxicity Assay Kit (Invitrogen™, Waltham, MA, USA) was used according to the manufacturer’s instructions. LDH activity was determined in the supernatant by measuring absorbance at 490 nm and subtracting the background absorbance at 680 nm using SpectraMax^®^ ABS (Molecular Devices, San Jose, CA, USA). The percentage of LDH leakage was calculated using the following formula:$$\mathrm{LDH}\;\mathrm{leakage}\;\left(\%\right)=\frac{\mathrm{LDH}\;\mathrm{activity}\;\mathrm{in}\;\mathrm{compound}-\mathrm{treated}\;\mathrm{samples}}{\mathrm{LDH}\;\mathrm{activity}\;\mathrm{in}\;\mathrm{vehicle}-\mathrm{treated}\;\mathrm{controls}}\;\times \;100$$

Data analysis was performed using SoftMax Pro 7.1.2 software (Molecular Devices).

### 2.6. Ig Detection

For the evaluation of Ig release, cells were centrifuged at 25 °C at 3000 rpm for 5 min; supernatants were collected and stored at −20 °C. IgM and IgG release were determined using an in-house assembled ELISA, using individual reagents from Sigma-Aldrich. 100 µL of anti-human IgG (Cat. No. I1886) and/or anti-human IgM (Cat. No. I0104) solutions at 1 µg/mL in PBS were plated in a 96-well plate and incubated overnight at 4 °C. Then, 100 µL of standards (0–1000 ng/mL, IgG from human serum Cat. No. I4506, IgM from human serum I8260) or sample (diluted or not in reagent buffer (PBS + 0.5% of bovine serum albumin + 0.05% of Tween 20)) were added to the plate and incubated for two hours at room temperature. After washing, 100 µL of anti-human polyvalent Igs (Cat. No. A3313) diluted 1:5000 in reagent buffer was added and incubated for 1 h at room temperature. As the last step, 100 µL of Phosphatase substrate 4 mg/mL (CAS # 333338-18-4, Sigma-Aldrich) diluted in alkaline phosphatase buffer (AP buffer) (composed of Tris CAS # 77-86-1, NaCl CAS # 7647-14-5, MgCl_2_•6H_2_O CAS # 7791-18-6, NaOH CAS # 1310-73-2, and H_2_O, all purchased by Sigma-Aldrich) was added. Absorbance was read at 405 nm, and the data were analyzed using the software SoftMax Pro 7.1.2. Results are expressed as fold change of chemical-treated cells versus vehicle-treated cells (DMSO, 0 µg/mL). Statistical analysis was performed using two-way ANOVA, followed by Dunnett’s test for BPA and analogs vs. DMSO (vehicle used as negative control, 0 µg/mL), and unpaired *t*-test with Welch correction for Ctrl + (Rapamycin 2 ng/mL) vs. DMSO (0 µg/mL) and male vs. female at the same concentration. Results were considered significant if *p* ≤ 0.05, with * and # *p* < 0.05, ** *p* < 0.01 vs. Ctrl.

### 2.7. Fate and Distribution In Vitro and Kinetic Models

The mathematical model In Vitro Mass Balance Equilibrium Partitioning Model version 2.0 (IV-MBM EQP v2.0) was used to estimate the intracellular concentrations of BPA and its analogs from the nominal concentrations tested in vitro [44,45,46]. These models consider the distribution of chemicals within the well, including binding to plastic, evaporation, and interaction with components of the medium, to predict actual cell exposure. However, this model only simulates neutral chemical forms and requires that parameters relating to the chemicals, cells and experiments had to be parameters. The chemical properties of BPA and its analogs were obtained from PubChem and are reported in Table 2.

The cells were characterized by specific biochemical and physical properties. The content of storage lipids was 0.5%, while membrane lipids accounted for 2.5% of tissue volume. Structural proteins, specifically non-lipid organic matter, represented 0.10 of the cellular composition. Cell density was 1 kg/L, and the system pH was maintained at 7.4. These values do not describe the specific composition of the PBMCs used in this study; rather, they reflect the general cellular modeling parameters commonly applied in IV-MBM EQP modeling approaches, as also done by Corsini et al. [38] for the same cell type. The cells were cultured in RPMI-1640 medium supplemented with the necessary nutrients and 5% d-FBS, as reported in the section above. As set up from the in vitro protocol used, the following parameters were used: 48-well plate with a well volume of 500 µL, average cell yield (seeding density) 630,000 cells, cell mass 3.15 ng, 5% d-FBS (albumin 24 g/L; lipids 1.9 g/L).

## 3. Results

Preliminary experiments were conducted to determine the highest non-cytotoxic concentration to be used for each compound. The CV80 after 24 h of exposure was calculated from dose–response experiments, as assessed by propidium iodide (PI) staining and flow cytometric analysis. To determine the CV80, analysis was conducted on two male and two female donors. The results are reported in Appendix A. In Table 3, the selected highest concentrations tested (CV80) and the range of concentrations used are listed. For BPS-MAE, a sex difference in the CV80 was observed, with males exhibiting greater sensitivity. Therefore, for this analog, a different range of concentrations was used.

### 3.1. Effects of BPA and BPA Analogs on Ig Release

The effects of BPA and its analogs on IgG and IgM release were assessed by ELISA in PBMCs stimulated with ODN2006 and rhIL-2 as described by Tuijnenburg et al. [37]. The protocol described by Tuijnenburg et al. [37] enables a robust B-cell activation and differentiation, where B cells are activated with CpG (a TLR9 ligand) in the presence of low IL-2, resulting in their differentiation into immunoglobulin-producing plasmablasts and secretion of IgG, IgM, and IgA after 6 days. The concentrations employed, as reported in Table 1, fall within the micromolar (corresponding to CV80 concentrations) and nanomolar (reflecting human exposure levels) ranges, which are considered relevant for simulating human exposure to these compounds [40,41]. PBMCs were treated for 24 h with increasing concentrations of BPA and its analogs (see Table 3), then stimulated with ODN2006 and rhIL-2 to induce Ig production for an additional 6 days. Rapamycin, a mTOR inhibitor known to inhibit Ig production, was used as a positive control (Ctrl +). At baseline, unstimulated (naïve) PBMCs produced mean IgG levels of 906.4 ± a standard error of the mean (SEM) of 198.6 ng/mL in male donors and 792.0 ± 312.8 ng/mL in female donors. Following stimulation with ODN2006 + rhIL-2 and the treatment with the vehicle control (DMSO), there was a significant increase in IgG production, reaching 80,262.8 ± 32,726.2 ng/mL in males and 66,100.7 ng/mL ± 21,560.1 in females. Similarly, baseline IgM levels in unstimulated cells were 54.1 ± 52.2 ng/mL in males and 14.6 ± 11.3 ng/mL in females, increasing to 57,929.5 ± 21,804.5 ng/mL and 1176.9 ± 666.3 ng/mL, respectively, upon stimulation. These data illustrate substantial inter-individual and sex-based variability in basal antibody production. For this reason, and to allow clearer visualization of compound-specific effects while preserving individual donor trends, the results are presented as stimulation indices normalized to each donor’s own control. This approach enables better comparability while maintaining consistency in the treatment-induced effects observed across donors. In Figure 1 and Figure 2, the results of IgG (Figure 1) and IgM (Figure 2) release, expressed as stimulation index (SI) values relative to vehicle-treated cells, are shown. The blue line corresponds to values obtained with male donors, whereas the red line represents those obtained with female donors.

The results show that BPA and its analogs can decrease IgG production (Figure 1). In general, all seven of the chemicals tested were able to reduce IgG release in a concentration-dependent manner in both male and female donors, except for BPZ, for which the decrease did not reach statistical significance. The suppression reached statistical significance at the highest concentrations tested (>1 µM). In the nM range, no suppression was observed. Conversely, a statistically significant increase was observed in male donors for BPAP at 15 nM (Figure 1B). Similar trends could be observed with BPE, BPP, and BPZ, but without reaching statistical significance. This effect was not observed in female donors, where only a significant reduction was found at the highest concentrations. Sex differences in IgG release can be observed, even if without statistical significance, except for BPAP at 0.015 µM (Figure 1B), which shows higher responsiveness to the tested substances in the female population. Ctrl + (represented by the black dot) reduced IgG release in all experiments, confirming the validity of the results.

Regarding IgM release (Figure 2), all compounds induced a concentration-dependent decrease, which became statistically significant at the highest tested concentrations. For many of the compounds tested, the level of suppression was comparable to that observed with the Ctrl + treatment. Notably, TCBPA caused a statistically significant decrease even at 2.5 nM in female donors (Figure 2G). Overall, the dose–response curves exhibited a similar trend across both male and female donors, indicating no substantial sex-based differences in the pattern of response.

To rule out cell death as the cause of the observed decrease in Ig release, an LDH release assay was conducted in parallel on PBMCs obtained from five male and five female donors. The results are reported in Appendix A. No cytotoxicity was observed, indicating that the decrease was not simply due to cell death.

To assess potential differences in potency, the concentrations required to inhibit 20% (IC_20_) or 50% (IC_50_) of the response compared to the control were calculated. The IC_20_ threshold was selected because it represents a low-effect concentration, enabling the identification of immunotoxic responses while avoiding overt cytotoxicity. This approach is consistent with current toxicological risk assessment practices aimed at evaluating subtle yet biologically relevant effects. The results are shown in Table 4. Overall, the IC_20_ and IC_50_ values were similar in male and female donors, indicating no substantial sex-based differences in the response pattern. However, the tested compounds show different IC values, indicating different potencies. The following ranking can be made based on the IC_20_ value: BPP > BPZ > BPAP > BPS-MAE > TCBPA > BPA > BPE. Overall, the results do not support the notion that BPA analogs are safer than BPA.

### 3.2. In Vitro Distribution Analysis

To determine the actual concentrations of BPA and its analogs entering our in vitro cell systems, we employed the IV-MBM EQP v2.0 developed by Armitage et al. [45,46]. This in silico model estimates the distribution of chemicals across the different compartments of the in vitro system, such as the medium, the plastic (specifically the multiwell culture plates used for treatments), the headspace, and the cells, by translating nominal concentrations into predicted intracellular concentrations. By accounting for processes such as sorption to plastic, volatilization, and partitioning into cellular and extracellular components, the model enables biologically relevant exposure levels to be defined more precisely. These data enable a more accurate assessment of the effective cellular exposure to bisphenols, thereby enhancing the interpretation of biological outcomes derived from in vitro assays.

The results are shown in Table 5. Interestingly, the cellular distribution (MF_cells_, highlighted in the pink column) varied significantly among the different bisphenols. BPP demonstrated the greatest cellular uptake (70.3%), with TCBPA closely behind (69.9%). Meanwhile, BPS-MAE and BPE exhibited the lowest values at 16.2% and 17.8%, respectively. When ranked by their fraction distributed within cells, the bisphenols are ordered as follows: BPP > TCBPA > BPZ > BPAP > BPA > BPE > BPS-MAE. This indicates that BPP and TCBPA strongly accumulate in cells, with over 69% of their fraction localized intracellularly, reflecting high affinity or uptake capacity. In contrast, bisphenols such as BPS-MAE and BPE exhibited significantly lower intracellular distribution (below 18%), suggesting that they either remained in the extracellular compartment or were bound to substances such as albumin or plastic. It is important to note that all bisphenols bind to plastic to varying degrees. These findings are crucial for understanding the actual cellular exposure to different bisphenols and may impact the assessment of their biological activity and toxicity in in vitro systems.

In order to investigate the relationship between bisphenol-induced immunosuppression and intracellular bioavailability, we examined the correlation between MF_cells_ and the inhibitory concentrations required to reduce ODN2006-stimulated Ig release. Figure 3 shows the relationship between MF_cells_ and IC_20_ values for IgG (Panel A) and IgM (Panel B) across various bisphenol analogs. Both panels demonstrate a consistent inverse correlation, indicating that compounds with higher cellular penetration capacity (i.e., increased MF_cells_) exhibit enhanced immunosuppressive potency and require lower concentrations to achieve equivalent inhibitory effects. While the correlation coefficients indicate a moderate negative relationship (IgG: Pearson r = −0.5898, Spearman r = −0.5509; IgM: Pearson r = −0.4265, Spearman r = −0.3114), none are statistically significant. This suggests that, although the trend is consistent, the association within this dataset is not statistically robust. Notably, the most lipophilic compounds, BPP and TCBPA, which have the highest MF_cells_ values (70.3% and 69.9%, respectively), have the lowest IC_20_ concentrations for both Igs. This pattern strongly supports the hypothesis that intracellular availability is a critical determinant of immunotoxic potency, with chemical structure influencing cellular penetration capacity and subsequent biological effects. These findings imply that the pharmacokinetic properties governing cellular uptake are as important as toxicodynamic mechanisms in determining the immunosuppressive potential of bisphenol analogs.

## 4. Discussion

The aim of this study was to investigate the effect of BPA and six related chemicals on human antibody production in vitro. All compounds, at non-cytotoxic concentrations, suppressed antibody release in a concentration-dependent manner, with differences in response. BPP showed the greatest potency, whereas BPE exhibited the least. Overall, the results indicate that BPA analogs suppress antibody production to varying degrees and do not appear to be safer than BPA. According to the results obtained in the IC_20_, we can establish a potency order (BPP > BPZ > BPAP > BPS-MAE > TCBPA > BPA > BPE).

Although both male and female PBMCs were included, and the general dose–response patterns were similar, some notable sex-specific differences emerged. A significant increase in IgG secretion was observed in male donors exposed to BPAP at 15 nM, within the range of human exposure, which was not seen in females. Conversely, females appeared to be slightly more sensitive to compounds such as BPP, BPE, and BPZ at higher concentrations, although statistical significance was only reached at µM levels. A particularly sensitive response to TCBPA was seen in female donors for IgM, with significant suppression occurring at 2.5 nM. This suggests that females are more responsive to some analogs at low doses. These findings are consistent with the idea that gender hormones and immune system regulation differ between males and females, and imply that EDs such as BPA analogs may have gender-specific immunomodulatory effects. While our data do not permit definitive conclusions regarding the underlying mechanisms, they emphasize the importance of considering gender as a biological variable in immunotoxicology studies.

The immunomodulatory effects of BPA and its analogs involve a complex interplay of pro-inflammatory and immunosuppressive mechanisms that are not fully understood. These mechanisms have significant implications for vaccine efficacy and allergic or autoimmune pathologies. A synthesis of the current evidence, combined with our own results, shows that the BPA actions are context-dependent and are influenced by the timing and dosage of exposure, the specific immune cell populations involved, and gender. Based on our results, we cannot conclude that BPA analogs currently used as substitutes have a lesser impact on the immune system than BPA itself. Currently, limited data on these analogs are available in the literature. Therefore, when discussing the results, we will primarily refer to BPA, for which data exists, while implying that the analogs may exert similar effects, given our preliminary findings.

For example, BPA enhances IgE production by activating the Ca^2+^/calcineurin/NF-AT pathway in CD4^+^ T cells, thereby driving IL-4 secretion and promoting B-cell differentiation towards an IgE^+^ phenotype [54]. This is consistent with epidemiological studies that have linked exposure to BPA to elevated levels of IgE and allergic sensitization. Conversely, BPA suppresses antigen-specific IgG responses, as demonstrated by lower seroconversion rates following HBV in individuals with higher urinary BPA levels [33]. This dichotomy highlights BPA’s ability to exacerbate Th2-driven inflammation while impairing adaptive humoral immunity, a paradox that may be explained by its contrasting effects on regulatory B cells (Bregs) and DCs. Specifically, BPA increases IL-4, leading to Breg cell production [54] and downregulates HLA-DR and CD86 on DCs [55]. This impairs T-cell priming and shifts the balance towards unregulated inflammation [54,55,56]. Moreover, the critical link between ER-AR and the immune system is that the ER promotes cytokine production and regulates cell differentiation [57], while AR regulates the development and activation of T and B lymphocytes [58]. BPA exhibits estrogenic activity, although its binding affinity for ER is 1000 to 10,000 times weaker than that of the physiological ligand 17β-estradiol [59]. The interaction of BPA with ER and AR modulates the expression of pro-inflammatory cytokines, such as TNF-α and IL-6, via NF-κB-dependent pathways. This creates a microenvironment favorable to autoimmune reactions [40,41,42,43,44,45,46,47,48,49,50,51,52,53,54,55,56,57,58,59,60]. BPA is also able to activate non-canonical receptors such as the aryl hydrocarbon receptor (AhR) [61] and the peroxisome proliferator-activated receptor (PPAR) [62]. The AhR regulates the expression of several genes, including those relevant to the immune cell subsets. It has been found in dendritic cells (DCs), macrophages, and lymphoid cells [63]. BPA can act as an AhR antagonist, and it downregulates AhR activity [64]. Concerning PPAR, it is known that it is involved in the alteration of the immune responses, and it has been found in macrophages [65], DCs [66], T [67], and B cells [68], and the interaction with BPA promotes the onset of inflammatory processes [62]. In line with our results, the suppression of protective Ig responses appears to be mediated by both nuclear and membrane ERs. These pathways are known to influence B-cell function, DC activation, and cytokine production, all of which are relevant to antibody generation. Prenatal BPA exposure alters the ERα/ERβ ratio, and ERβ overexpression in fish macrophages has been linked to reduced adaptive immunity [69]. Furthermore, in vitro studies demonstrate that BPA binds to GPER (a membrane-associated ER) to inhibit IL-10 and IL-13 secretion in human male lymphocytes. These two cytokines are critical for B-cell maturation and antibody class switching [56] and may lead to a reduction in IgG and IgM production, as observed in our results. Taken together, these findings suggest that BPA disrupts Ig production via both genomic and non-genomic pathways, potentially involving crosstalk with MAPK or NF-κB [69]. Although this study evaluated only the functional outcome of the immunotoxic potential of BPA and its analogs through antibody secretion, it did not directly assess the molecular pathways involved. The absence of mechanistic endpoints, such as receptor-binding assays, gene expression profiling, or pathway-specific markers, limits the ability to draw definitive conclusions about the underlying mechanisms. Nevertheless, the existing literature reporting above indicates that bisphenols may exert their effects through various pathways that are known to influence B-cell function as well. However, the integration of these mechanisms remains hypothetical in the context of the present data and warrants further investigation that can clarify the molecular events that mediate antibody suppression.

The ability of BPA to impair vaccine-induced immunity has significant implications for public health, given its widespread presence in consumer products and the environment. A large cross-sectional study based on the National Health and Nutrition Examination Survey (NHANES) demonstrated that higher urinary BPA concentrations were significantly correlated with an increased susceptibility to HBV infection despite vaccination [33]. This suggests that exposure to BPA may reduce the protective antibody response triggered by HBV, which could compromise herd immunity and increase the risk of viral transmission in populations with a high BPA burden [33]. Mechanistically, BPA exerts immunomodulatory effects that could underlie this impaired vaccine responsiveness. It induces apoptosis and necrosis in human monocytes, reducing their viability and leading to a decrease in the number of DCs and their impaired function. DCs are crucial for the uptake, processing, and presentation of antigens to T cells. BPA also reduces the endocytic capacity of DCs for vaccine antigens such as Hepatitis B surface antigen (HBsAg), thereby hindering the initiation of adaptive immune responses [33]. Furthermore, BPA exposure increases the production of reactive oxygen species in human B lymphocytes, which may reduce the pool of HBsAg-specific memory B cells that are essential for durable humoral immunity [33]. Interestingly, the association between BPA and reduced vaccine efficacy was inconsistent across NHANES survey cycles, showing a downward trend over time [33]. This may be due to declining BPA exposure resulting from regulatory efforts and public awareness campaigns since the early 2000s. However, it may also be influenced by confounding factors such as age and birth cohort effects that impact immune responses. Nevertheless, these findings highlight the importance of continued surveillance of BPA exposure and its immunological consequences.

In the context of immune function, research on the T cell-dependent antibody response (TDAR), widely recognized as the benchmark for evaluating adaptive humoral immunity, is somewhat limited but still provides valuable insights. One murine study reported that developmental BPA exposure modulated innate immunity without impairing the antiviral adaptive response, including virus clearance and presumably antibody production [70]. However, this finding contrasts with epidemiological data suggesting impaired vaccine responses in humans, highlighting species differences and the complexity of BPA’s immunomodulation. Emerging research on trained immunity suggests that BPA can induce epigenetic and metabolic reprogramming in human monocytes, which could alter cytokine production profiles upon secondary stimulation [40]. Although trained immunity primarily involves innate immune cells, modulation by BPA could indirectly also influence the adaptive responses, including antibody generation.

Taken together, these findings and our results imply that exposure to BPA and its analogs can compromise vaccine-induced humoral immunity by exerting direct cytotoxic effects on antigen-presenting cells and B lymphocytes, interfering with the processing and presentation of antigens, and triggering oxidative stress mechanisms. This has critical implications for vaccination programs, particularly in vulnerable populations such as children and immunocompromised individuals, who may already have suboptimal immune responses.

Moreover, a significant challenge in comparing different studies is the non-monotonic dose–response curve of BPA. For example, low doses of BPA (0.1–1 nM) suppress DCs activation, whereas higher doses (10–100 nM) can paradoxically enhance the release of inflammatory cytokines [56]. Furthermore, structural analogs such as BPS and BPAF exhibit different immunotoxic profiles. BPS amplifies pro-inflammatory CD86^+^ B cells, whereas BPAF broadly suppresses Th2 cytokines [56]. Although we did not observe differences among the various analogs in their effects on IgG and IgM release in our systems, causing a general immunosuppression, it is important to acknowledge that the analogs employed have different potencies, resulting in immunosuppressive effects at varying concentrations. This variability highlights the inadequacy of assuming functional equivalence among all the BPA substitutes and emphasizes the need for analog-specific risk assessments.

When assessing the potential hazard of BPA and its analogs, it is also essential to consider their molecular structures and the substituents incorporated into new compounds. Our findings show that alterations to the structure of BPA analogs affect two important aspects of their biological activity: cellular pharmacokinetics (uptake) and pharmacodynamics (intrinsic potency). When considering the structural diversity of these analogs, they can be broadly categorized according to the nature of their central bridging groups and the presence of specific substituents on the aromatic rings. For instance, bisphenols with aliphatic bridges, such as BPA, BPE, and BPP, mainly differ in the length and branching of their central linkers. Others, such as BPAP, contain aromatic bridges, while BPZ is characterized by a cyclic bridge. Additionally, some analogs incorporate halogenated substituents, as exemplified by the chlorinated TCBPA, or sulfonic bridges as in BPS-MAE. This structural classification is not merely descriptive, but it reflects meaningful differences that can significantly influence the chemical behavior and biological activity of each compound. Understanding these distinctions is crucial for interpreting their toxicological profiles and for the rational development of bisphenol derivatives with improved safety profiles. The inverse relationship observed indicates that compounds with greater intracellular bioavailability seem to possess higher intrinsic toxicity, and vice versa. This distinction emphasizes the complexity of structure–activity relationships, where the molecular features governing membrane permeability and cellular accumulation may differ from those determining molecular target affinity and toxic effects. Therefore, evaluating both cellular uptake and potency is essential for a comprehensive understanding of the toxicological profiles of BPA analogs, as well as for guiding the design of safer bisphenol derivatives.

## 5. Conclusions

Health authorities prioritize reducing BPA exposure, particularly during critical early life stages, to protect health [15]. While regulatory attention has focused on BPA, it is crucial to recognize that there is currently a very limited risk assessment for BPA analogs. Our study demonstrates that these substitutes do not necessarily offer a safer immunological profile compared to BPA itself, as they can also modulate immune responses. Therefore, regulatory frameworks and research efforts should broaden their scope to include these analogs to fully address the immunotoxic risks posed by bisphenols. Moreover, prospective longitudinal studies are urgently needed to clarify the impact of BPA and its analogs on antibody production, infections, and vaccine responsiveness over time. These studies will be essential to identify potential interventions capable of mitigating bisphenol-induced immunotoxicity and safeguarding public health.

An important consideration that was not addressed in our study is also the simultaneous exposure to BPA, its analogs, and other EDs from multiple environmental sources. For instance, Sonavane et al. [71] reviewed numerous co-exposure studies that highlighted the combined effects of BPA with various environmental contaminants, natural compounds, or therapeutic agents. Notably, co-exposure to BPA and perfluorinated compounds such as perfluorooctanoic acid and perfluorooctanesulfonic acid, widely used industrial chemicals, has been shown to impair myocardial differentiation in vitro [72]. This exemplifies the complex, multifactorial nature of real-world chemical exposures that may exacerbate immunotoxic effects beyond those observed with BPA alone. In conclusion, a comprehensive approach that considers combined exposure to BPA and its analogs, as well as other environmental contaminants, is also essential for accurate risk assessment and effective public health interventions. Only through such integrated strategies can we ensure the protection of immune function and vaccine efficacy in populations worldwide.

## Figures and Tables

**Figure 1 life-15-01203-f001:**
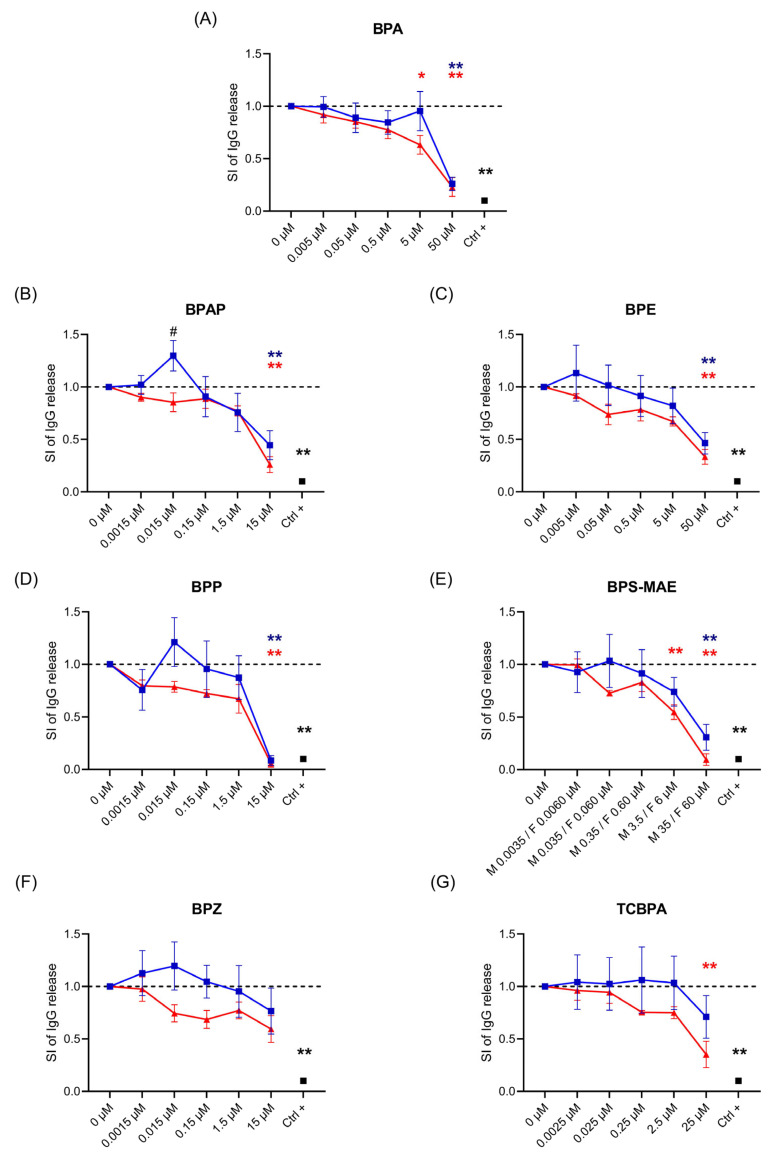
Effect of BPA and BPA analogs on IgG production. BPA (**A**), BPAP (**B**), BPE (**C**), BPP (**D**), BPS-MAE (**E**), BPZ (**F**), and TCBPA (**G**), Ctrl + and DMSO (0 µM). Cells were exposed to increasing concentrations of chemicals for 24 h, then stimulated with ODN2006 and rhIL-2 for 6 days. The results are expressed as SI of IgG compared to the vehicle DMSO (0 µM). Each dot represents the mean ± SEM, with n = 5 female (red) and 5 male (blue) donors. Statistical analysis was performed by two-way ANOVA, followed by Dunnett’s test for BPA and analogs vs. DMSO (0 µM). For comparisons between the Ctrl + (Rapamycin 2 ng/mL—black) and DMSO (0 µM), an unpaired *t*-test with Welch correction was applied. * indicates significance relative to DMSO (blue for males, red for females in figures), while # indicates significant differences between males and females at the same concentration. Results were considered statistically significant at *p* ≤ 0.05, with * *p* < 0.05, ** *p* < 0.01, and # *p* < 0.05, vs. Ctrl.

**Figure 2 life-15-01203-f002:**
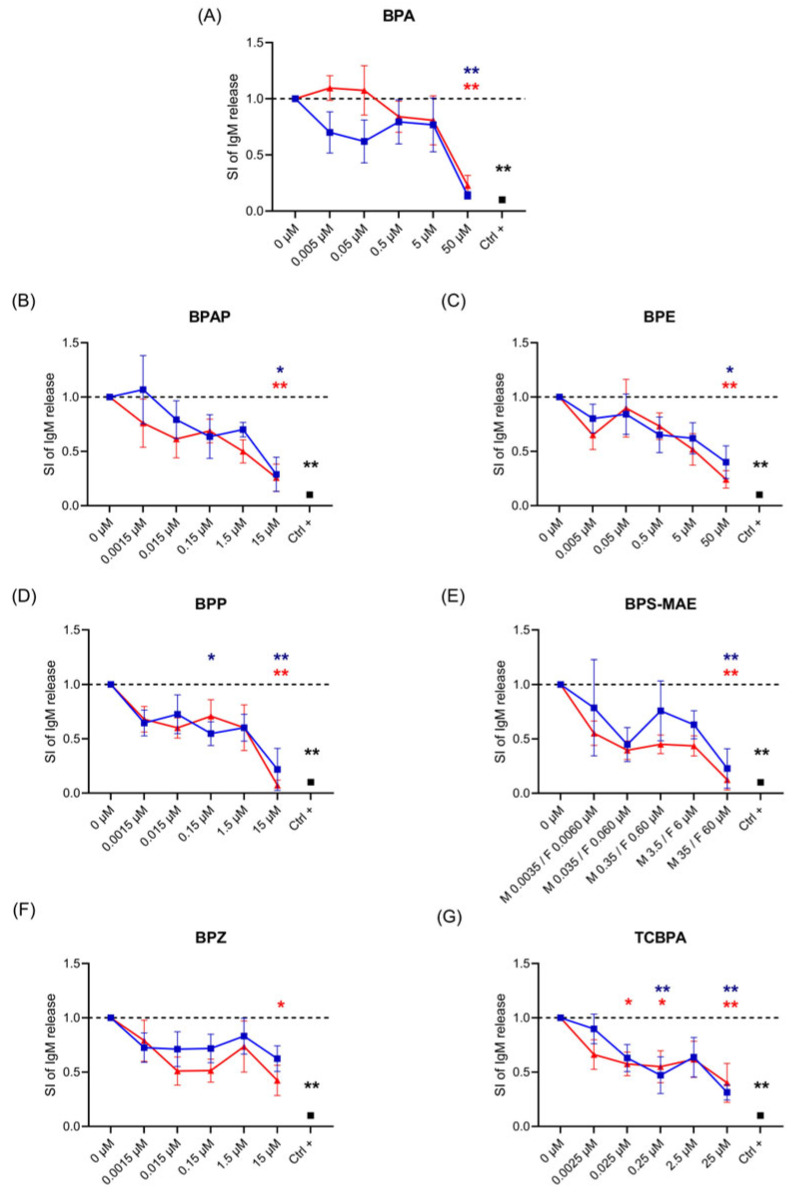
Effect of BPA and BPA analogs on IgM production. (**A**) BPAP, (**B**) BPE, (**C**) BPP, (**D**) BPS-MAE, (**E**) BPZ, (**F**) TCBPA, and (**G**) Ctrl + and DMSO (0 µM). Cells were exposed to increasing concentrations of chemicals for 24 h, then stimulated with ODN2006 and rhIL-2 for 6 days. The results are expressed as SI of IgM compared to the vehicle DMSO (0 µM). Each dot represents the mean ± SEM, with n = 5 female (red) and 5 male (blue) donors. Statistical analysis was performed by two-way ANOVA, followed by Dunnett’s test for BPA and analogs vs. DMSO (0 µM), and unpaired *t*-test with Welch correction for Ctrl + (Rapamycin 2 ng/mL—black) vs. DMSO (0 µM). Results were considered significant if *p* ≤ 0.05, with * *p* < 0.05, ** *p* < 0.01 vs. Ctrl. The color of the asterisks corresponds to the gender, with blue representing males and red representing females in the figures.

**Figure 3 life-15-01203-f003:**
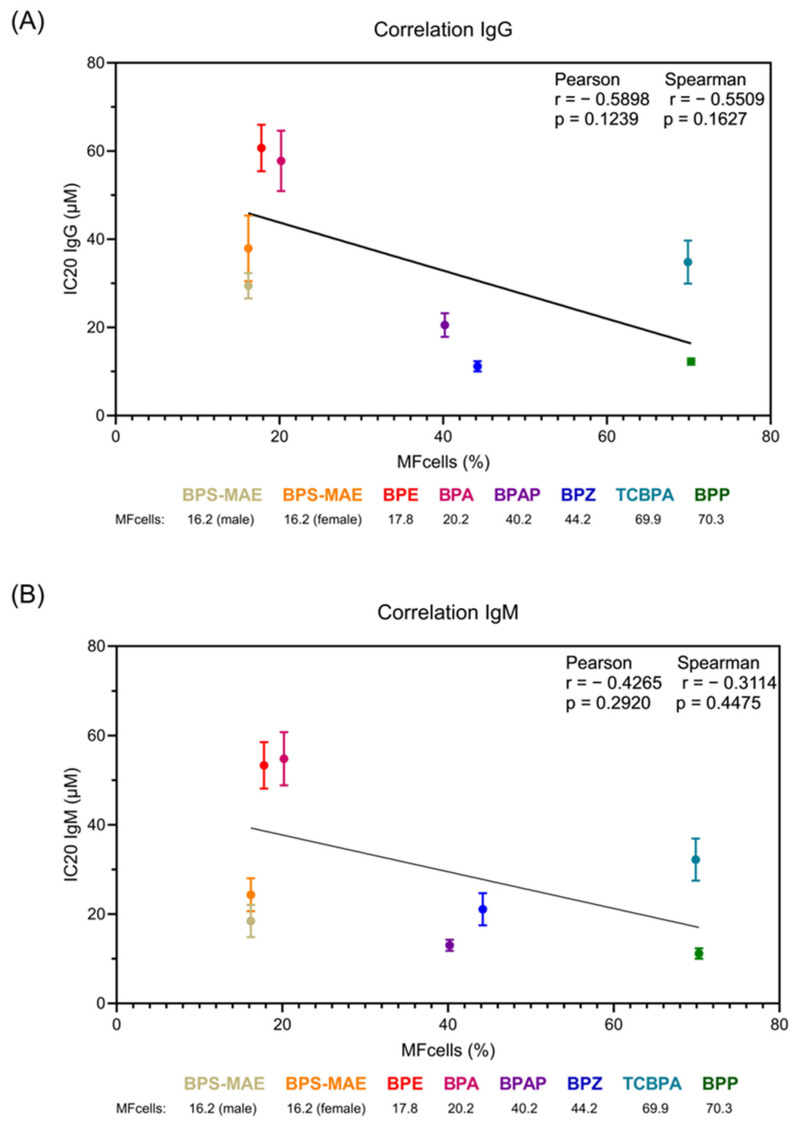
Correlation between MF_Cells_ (%) and IC_20_ following exposure to BPA and its analogs. (**A**) Correlation between IC_20_ IgG results and MF_Cells_. (**B**) Correlation between IC_20_ IgM results and MF_Cells_.

**Table 1 life-15-01203-t001:** Name, acronym, CAS number, chemical structure, molecular weight (MW), and logP of the tested compounds.

Name	Acronym	CAS N°	Chemical Structure
Bisphenol A	BPA	80-05-7	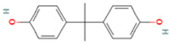
Bisphenol AP	BPAP	1571-75-1	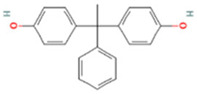
Bisphenol E	BPE	2081-08-5	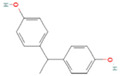
Bisphenol P	BPP	2167-51-3	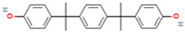
Bisphenol S 4-allyl ether	BPS-MAE	97042-18-7	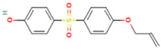
Bisphenol Z	BPZ	843-55-0	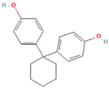
3,3′,5,5′-Tetrachlorobisphenol A	TCBPA	79-95-8	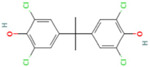

**Table 2 life-15-01203-t002:** Physicochemical properties collected from PUBCHEM.

Name	MW (g/mol)	MP (°C)	IOC Type	pKa	log K_OW,N_	log K_AW,N_	C_SAT,W,N_ (mg/L)	ECx in μM	Reference
BPA	228.3	158.0	A	9.60	3.32	−9.43	3.80 × 10^2^	50.00	[47]
BPAP	290.4	189.0	A	10.22	4.86	−10.64	1.20 × 10^2^	15.00	[48]
BPE	214.3	125.0	A	10.10	3.19	−9.55	2.50 × 10^3^	50.00	[49]
BPP	346.5	195.0	A	10.08	6.25	−10.23	5.00 × 10	15.00	[50]
BPZ	268.4	189.0	A	9.91	5.00	−9.41	1.00 × 10^2^	15.00	[51]
TCBPA	366.1	136.0	A	6.91	6.22	−8.50	4.00 × 10	25.00	[52]
BPS-MAE (M)	290.3	172.0	A	8.20	3.10	−9.00	1.50 × 10^3^	35.00	[53]
BPS-MAE (F)	290.3	172.0	A	8.20	3.10	−9.00	1.50 × 10^3^	60.00	[53]

Table legend: ECx in μM is the highest nominal concentration used in the in vitro test system. MW (g/mol): molecular weight, expressed in grams per mole; MP (°C): melting point, expressed in degrees Celsius; IOC Type: ionization/organic chemical type, indicating the chemical classification based on ionization state or chemical nature; pKa: acid dissociation constant, representing the pH at which the compound is 50% ionized; log K_OW,N_: logarithm of the octanol-water partition coefficient for the neutral form, indicating hydrophobicity; log K_AW,N_: logarithm of the air-water partition coefficient for the neutral form, indicating volatility; C_SAT,W,N_ (mg/L): saturation concentration in water for the neutral form, expressed in milligrams per liter. M: male; F: female.

**Table 3 life-15-01203-t003:** CV80 of BPA and its analogs determined by PI staining.

Name	CV80 Male (µM)	CV80 Female (µM)	Selected Highest Concentration (µM)	Concentrations Tested (µM)
BPA	50.84 ± 4.3	54.37 ± 6.1	50	0.005–0.05–0.5–5–50
BPAP	12.90 ± 3.9	15.74 ± 1.2	15	0.0015–0.015–0.15–1.5–15
BPE	51.87 ± 9.1	59.33 ± 2.7	50	0.005–0.05–0.5–5–50
BPP	13.13 ± 16.0	20.44 ± 2.0	15	0.0015–0.015–0.15–1.5–15
BPS-MAE *	35.80 ± 3.4	60.65 ± 2.9	35 (M^#^) 60 (F^##^)	0.0035–0.035–0.35–3.5–35 (M) 0.0060–0.060–0.60–6–60 (F)
BPZ	13.81 ± 3.6	17.82 ± 1.2	15	0.0015–0.015–0.15–1.5–15
TCBPA	17.60 ± 8.5	30.23 ± 0.1	25	0.0025–0.025–0.25–2.5–25

Each value of CV80 represents the mean ± SEM, with *n* = 2 female (F) and 2 male (M) donors. Statistical analysis was performed by unpaired *t*-test with Welch’s correction to assess significant differences between CV80 values obtained in males and females for the same substance. Results were considered statistically significant at *p* ≤ 0.05, with * *p* < 0.05. Legend: M^#^: male F^##^: female.

**Table 4 life-15-01203-t004:** IC_50_ and IC_20_ values of BPA and its analogs for the inhibition of IgG and IgM secretion. Results are stratified by gender, and to highlight potential sex-related differences, statistical analysis was performed using BPA as a reference, the lead compound.

**IC_50_ (µM)**				
	**IgG**		**IgM**	
	**Male**	**Female**	**Male**	**Female**
BPA	33.1 ± 6.1	29.2 ± 6.8	25.0 ± 4.3	32.0 ± 6.9
BPAP	14.9 ± 3.4 *	9.8 ± 1.4 *	7.2 ± 0.7 *	13.8 ± 6.7
BPE	52.3 ± 14.7	34.9 ± 6.5	23.2 ± 5.0	35.9 ± 12.0
BPP	8.5 ± 1.4 *	5.8 ± 0.7 *	4.9 ± 0.3 **	5.80 ± 1.1 *
BPS-MAE	27.5 ± 6.7	14.8 ± 2.3	16.9 ± 3.4	22.0 ± 9.6
BPZ	15.3 ± 4.7	12.6 ± 3.0	7.9 ± 6.4	15.6 ± 9.5
TCBPA	27.5 ± 9.8	15.6 ± 2.2	20.0 ± 4.5	14.3 ± 11.7
**IC_20_ (µM)**				
	**IgG**		**IgM**	
	**Male**	**Female**	**Male**	**Female**
BPA	59.7 ± 10.1	55.9 ± 10.3	53.5 ± 7.4	56.2 ± 10.3
BPAP	23.1 ± 4.5 **	18.0 ± 3.0 *	13.7 ± 0.7 **	12.4 ± 2.5 *
BPE	60.4 ± 5.6	60.9 ± 9.1	55.4 ± 10.1	51.8 ± 6.5
BPP	12.5 ± 1.3 **	12.1 ± 0.9 *	11.1 ± 0.6 **	11.2 ± 2.0 *
BPS-MAE	37.9 ± 7.4	29.5 ± 2.9	24.4 ± 3.7 *	18.5 ± 3.6 *
BPZ	11.1 ± 1.3 **	11.2 ± 2.0 *	20.0 ± 4.3 **	21.8 ± 9.8 *
TCBPA	42.3 ± 8.1	29.2 ± 5.1	35.5 ± 6.0	26.8 ± 8.0

Table legend: Each value represents the mean ± SEM, with *n* = 5 female and 5 male donors. Statistical analysis was performed by *t*-test with Welch’s correction for BPA analogs vs. BPA. * represents significance relative to BPA. Results were considered statistically significant at *p* ≤ 0.05, with * and *p* < 0.05 **.

**Table 5 life-15-01203-t005:** In vitro mass fraction distribution (%) of the different bisphenols.

Name	MF_BULK WAT_	MF_ALB_	MF_S-LIP_	MF_WAT_	MF_DOM_	MF_Cells_	MF_Plastic_
BPA	63.6%	24.7%	5.3%	33.6%	0.0%	**20.2%**	16.2%
BPAP	53.2%	41.0%	10.3%	1.8%	0.0%	**40.2%**	6.6%
BPE	65.8%	21.2%	4.7%	39.9%	0.0%	**17.8%**	16.4%
BPP	27.0%	9.3%	17.5%	0.1%	0.0%	**70.3%**	2.8%
BPZ	49.5%	36.7%	11.3%	1.5%	0.0%	**44.2%**	6.3%
BPS-MAE (M)	67.6%	18.9%	4.2%	44.5%	0.0%	**16.2%**	16.2%
BPS-MAE (F)	67.6%	18.9%	4.2%	44.5%	0.0%	**16.2%**	16.2%
TCBPA	27.3%	9.7%	17.5%	0.1%	0.0%	**69.9%**	2.8%

Table legend: Mass balance (%) was calculated using the model developed by Armitage et al. (2021) [46]. MF_ALB_, fraction bound to albumin; MF_S-LIP_, fraction bound to lipids; MF_DOM_, fraction bound to dissolved organic matter; MF_WAT_, fraction in water; MF_BULKWAT_ the some of the MF_ALB_, MF_S-LIP_, MF_WAT_, and MF_DOM_; MF_cells_, fraction into the cells; MF_Plastic_, fraction bound to plastic. M: male; F: female. Values in bold: highlight the most important result.

## Data Availability

The original contributions presented in the study are included in the article/Appendix A, further inquiries can be directed to the corresponding author.

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
