# Peer review of "Disrupting Defenses: Effects of Bisphenol A and Its Analogs on Human Antibody Production In Vitro"

_life, 2025, doi:10.3390/life15081203_

Round 1

Reviewer 1 Report

Comments and Suggestions for Authors

The manuscript addresses a highly relevant toxicological concern by investigating the immunotoxic effects of Bisphenol A (BPA) and six of its analogues on human antibody production. Using a robust in vitro model (PBMCs from both sexes), the authors explore IgM and IgG secretion in response to bisphenol exposure. The topic is timely, scientifically valuable, and fits well within the scope of Life. The study is methodologically sound and contains a high degree of detail, including quantitative modeling of intracellular concentrations-a notable strength.

However, some points need clarification and refinement to improve the clarity, transparency, and impact of the work.

Major Comments

1. Lack of mechanistic insight beyond antibody production

While IgG/IgM suppression is clearly demonstrated, the molecular mechanisms remain speculative.

2. Justification of Concentration Ranges and Relevance to Human Exposure

The manuscript refers to environmentally relevant concentrations but lacks a concise summary of exposure data from biomonitoring studies to support this. Consider adding a table or brief paragraph comparing used concentrations with those found in human blood/urine.

3. Sex Differences Not Fully Explored

Although both sexes were included and some differences were found, these results are under-discussed.

4. Statistical Treatment and Sample Size

The study uses five donors per sex, which is acceptable but somewhat limited for generalization. Also, ethical approve?

Mechanistic Limitations Acknowledged, but Could Be Strengthened

While the authors discuss ER/AR and AhR-mediated pathways, no direct mechanistic data were collected.

Minor Comments

  • Improve figure captions (particularly Figures 1–2 and S1) to include descriptions of statistical markers, lines, and symbols.

Comments on the Quality of English Language
  • The manuscript is generally well written but includes minor grammatical and typographic issues (e.g., “immosuppresion” instead of “immunosuppression”).
  • Define all abbreviations at first use in both main and supplementary documents (e.g., SI, Ctrl+).

Author Response

Comments and Suggestions for Authors: The manuscript addresses a highly relevant toxicological concern by investigating the immunotoxic effects of Bisphenol A (BPA) and six of its analogues on human antibody production. Using a robust in vitro model (PBMCs from both sexes), the authors explore IgM and IgG secretion in response to bisphenol exposure. The topic is timely, scientifically valuable, and fits well within the scope of Life. The study is methodologically sound and contains a high degree of detail, including quantitative modeling of intracellular concentrations-a notable strength.

However, some points need clarification and refinement to improve the clarity, transparency, and impact of the work.

Thank you very much for your thoughtful and constructive comments and suggestions. We greatly appreciate the time and effort you have invested in reviewing our manuscript. Below, we provide detailed responses addressing each point raised.

Major Comments

Comment 1: Lack of mechanistic insight beyond antibody production: while IgG/IgM suppression is clearly demonstrated, the molecular mechanisms remain speculative.
Response 1: Thank you for this important observation. As stated in the manuscript, the aim of the present study was to evaluate the immunotoxic potential of BPA and its analogues by assessing IgM and IgG production. While mechanistic pathways were beyond the scope of this investigation, we have proposed a plausible hypothesis based on existing literature, which should be further investigated in future studies specifically designed to elucidate these mechanisms. Unfortunately, current literature, particularly regarding BPA analogues, remains limited, and generating such mechanistic data will be essential to better understand their immunotoxic profile.

Comment 2: Justification of Concentration Ranges and Relevance to Human Exposure: the manuscript refers to environmentally relevant concentrations but lacks a concise summary of exposure data from biomonitoring studies to support this. Consider adding a table or brief paragraph comparing used concentrations with those found in human blood/urine.
Response 2: We appreciate this valuable suggestion. Unfortunately, there is currently a lack of comprehensive biomonitoring data for all the BPA analogues investigated, particularly in terms of their concentrations in human blood or urine. Nevertheless, we have added a supplementary paragraph summarizing available data for BPA and selected analogues (lines 45-51), and we clarified the rationale for the concentration ranges used in our experiments, which are within or close to the ranges considered environmentally relevant (lines 113-117).

Comment 3: Sex Differences Not Fully Explored: although both sexes were included and some differences were found, these results are under-discussed.
Response 3: Thank you for highlighting this point. We have now expanded the discussion on sex-related differences in immune response, referencing our findings and supporting literature (lines 418-431).

Comment 4: Statistical Treatment and Sample Size: the study uses five donors per sex, which is acceptable but somewhat limited for generalization. Also, ethical approve?
Response 4: Thank you for your comment. Current guidelines for in vitro studies recommend the use of at least five donors, regardless of sex, to account for inter-individual variability. In our study, we doubled this minimum number by including five male and five female donors. This allowed us not only to obtain robust and comparable data across individuals but also to identify potential sex-related differences in response, where present. Based on our experience and on practices commonly reported in the literature for studies using human PBMCs, we consider the sample size used to be appropriate and scientifically sound.

Regarding ethical approval, this study did not require formal approval by an ethics committee, as we used buffy coats obtained from healthy, anonymous donors through an established agreement with the Niguarda Hospital blood bank (Milan, Italy). The study involved exclusively in vitro exposure to substances, with no direct involvement of human subjects. This exempts it from ethical approval requirements, in accordance with the definition of "buffy coat" as a blood component not linked to clinical procedures (Directive 2004/33/EC, Art. 15) and with current national regulations on Ethics Committees (Ministerial Decrees of July 15, 1997, and March 18, 1998), which do not require approval for in vitro studies using such materials.

Comment 5: Mechanistic Limitations Acknowledged, but Could Be Strengthened. While the authors discuss ER/AR and AhR-mediated pathways, no direct mechanistic data were collected.
Response 5: As mentioned above, the primary objective of this work was not to elucidate molecular mechanisms, but rather to assess immunotoxic potential through antibody production. While we discussed potential involvement of ER/AR and AhR-mediated pathways based on literature, we have now reinforced this limitation in the discussion and emphasized the need for future mechanistic studies (lines 468-488).

Minor Comments

Improve figure captions (particularly Figures 1–2 and S1) to include descriptions of statistical markers, lines, and symbols. Thank you for the suggestion. We have revised all figure captions to include clear descriptions of statistical markers, lines, and symbols to enhance interpretability.

Comments on the Quality of English Language

The manuscript is generally well written but includes minor grammatical and typographic issues (e.g., “immosuppresion” instead of “immunosuppression”).
We appreciate the feedback. We have carefully proofread the manuscript and corrected all grammatical and typographical errors.

Define all abbreviations at first use in both main and supplementary documents (e.g., SI, Ctrl+).
All abbreviations have been reviewed.

Reviewer 2 Report

Comments and Suggestions for Authors

In the present manuscript "Disrupting defenses: effects of bisphenol A and its analogues on human antibody production in vitro" by Carlotta Passoni et al, the  immunotoxic effects of bisphenol A (BPA) and six BPA analogues on antibody production in human peripheral blood mononuclear cells (PBMCs) were assessed. Authors showed that BPA and its analogues suppress IgG and IgM production in a concentration-dependent manner, with BPP being the most potent and BPE the least. These findings indicate that BPA analogues are not actually safer than BPA and may pose similar or greater immunotoxic risks. The study contributes meaningfully to the field of immunotoxicology, particularly in the context of endocrine-disrupting chemicals (EDCs) and immunomodulators.
However, several minor areas of the manuscript could be revised to improve clarity:
1. Introduction
- When discussing previous studies on BPA and immunoglobulin production, please clarify which findings were obtained from in vivo models and which were derived from in vitro experiments to improve understanding of this part. 

2.Materials and Methods 
- Please clarify whether replicates were performed per donor or were pooled.
-  Consider justifying the use of IC20 threshold and its relevance to risk assessment.

Author Response

Comments and Suggestions for Authors: In the present manuscript "Disrupting defenses: effects of bisphenol A and its analogues on human antibody production in vitro" by Carlotta Passoni et al, the  immunotoxic effects of bisphenol A (BPA) and six BPA analogues on antibody production in human peripheral blood mononuclear cells (PBMCs) were assessed. Authors showed that BPA and its analogues suppress IgG and IgM production in a concentration-dependent manner, with BPP being the most potent and BPE the least. These findings indicate that BPA analogues are not actually safer than BPA and may pose similar or greater immunotoxic risks. The study contributes meaningfully to the field of immunotoxicology, particularly in the context of endocrine-disrupting chemicals (EDCs) and immunomodulators.

However, several minor areas of the manuscript could be revised to improve clarity.

Thank you very much for your thoughtful and constructive comments and suggestions. We greatly appreciate the time and effort you have invested in reviewing our manuscript. Below, we provide detailed responses addressing each point raised:

  1. Introduction

Comment 1: When discussing previous studies on BPA and immunoglobulin production, please clarify which findings were obtained from in vivo models and which were derived from in vitro experiments to improve understanding of this part.
Response 1: Thank you for your comment. We agree that this distinction is important. The relevant section of the Introduction (lines 67–78) already specifies whether the cited findings were obtained from in vivo or in vitro studies. We have now slightly revised the wording to ensure even greater clarity in differentiating the models used in the referenced literature.

2.Materials and Methods

Comment 2: Please clarify whether replicates were performed per donor or were pooled.
Response 2: We thank the reviewer for this observation. As stated in the figure legends, each data point in the graphs represents the mean ± SEM of five independent donors. Experiments were performed separately for each donor, and replicates were not pooled.

Comment 3: Consider justifying the use of IC20 threshold and its relevance to risk assessment.
Response 3: Thank you for the suggestion. A sentence has been added (line 337-341) to clarify that the IC20 threshold was chosen as it reflects a low-effect level, which is relevant for estimating potential immunotoxicity in realistic exposure scenarios without inducing overt cytotoxicity. This aligns with current approaches in toxicological risk assessment.

Reviewer 3 Report

Comments and Suggestions for Authors

The authors investigated the effects of bisphenol A and its analogs on IgG secretion. IgG is an extremely diverse molecule, each IgG molecule is specific to a particular antigen. IgG secretion should be considered in the context of some external influence, such as a viral infection. Estimation of the total amount of IgG has no biological meaning. Experiments on the secretion of total IgG by mononuclear cells in a test tube are far from the real picture. The Methods section requires significant revision. The figures are not clear and should be redone.
Also below are some comments:

1. Line 100. What is the human model of antibody secretion? Please clarify.
2. Paragraph “Igs detection”. This method raises doubts. Firstly, after the stage of antigen sorption in the wells of the plates, there is no blocking stage. This significantly affects the result obtained. Second, anti-human polyvalent Ig (Cat. No. A3313) can react with adsorbed antibodies to human IgG (Cat. No. I1886) and/or antibodies to human IgM (Cat. No. I0104).

3. Line 177. Please specify which mathematical models were used. How were these mathematical models applied?

4. Line 185. Please provide the reference and date of access to the PubChem resource

5. Line 199-207. “The content of storage lipids was 0.5%, while membrane lipids accounted for 2.5% of tissue volume. Structural proteins, specifically non-lipid organic matter, represented 0.10 of the cellular composition.” By what methods were these data obtained? What do they indicate and what is their significance in the context of this manuscript? These data do not characterize the mononuclear cells used in this study. A quantification of B cells that are involved in IgG secretion should be provided here.

6. Line 210-213. Please provide the method for propidium iodide (PI) staining and flow cytometry analysis in the Methods section. The flow cytometric results should also be provided and described in this manuscript. How was cell viability tested? Please also describe in the Methods section.

7. Line 237. What is the “stimulation index”? Please provide the antibody concentrations obtained in common units, such as μg/ml. The concentration of IgG produced by untreated cells should also be provided. Are there statistically significant differences between treated and untreated cells?

8. Line 286. How does lactate dehydrogenase demonstrate cell viability? Please explain. This should also be described in the Methods section

9. Line 307. What is the mathematical model used based on? How can the differences in cell penetration for different compounds be explained? How is this related to their structure?

10. Line 322. Please clarify what plastic we are talking about?

Author Response

Comments and Suggestions for Authors:  The authors investigated the effects of bisphenol A and its analogs on IgG secretion. IgG is an extremely diverse molecule, each IgG molecule is specific to a particular antigen. IgG secretion should be considered in the context of some external influence, such as a viral infection. Estimation of the total amount of IgG has no biological meaning. Experiments on the secretion of total IgG by mononuclear cells in a test tube are far from the real picture. The Methods section requires significant revision. The figures are not clear and should be redone. We respectfully acknowledge the reviewer’s point. Our goal was not to assess antigen-specific IgG responses but rather to evaluate whether BPA and its analogues exert general immunotoxic effects on antibody production. Measuring total IgG and IgM in vitro allows for the identification of potential immunomodulatory effects on B-cell function, independent of antigen specificity. This model, although simplified, is widely used in immunotoxicology to screen immunosuppressive compounds and investigate B-cell activity in controlled conditions. The experimental setup, based on the established protocol by Tuijnenburg et al. (2020), offers a reproducible platform for assessing human B-cell responses to chemical exposures.

Thank you very much for your thoughtful and constructive comments and suggestions. We greatly appreciate the time and effort you have invested in reviewing our manuscript. Below, we provide detailed responses addressing each point raised.

Also below are some comments:

Comment 1: Line 100. What is the human model of antibody secretion? Please clarify.
Response 1: Thank you for your question. The model employed is an in vitro differentiation model of human PBMCs that allows the induction of antibody-secreting cells, following stimulation. This is described in detail in the Materials and Methods section and is based on the protocol by Tuijnenburg et al. (2020), which uses a cocktail of IL-2 and CpG to promote B-cell differentiation and IgG/IgM secretion.
Reference: Tuijnenburg, P., Aan de Kerk, D. J., Jansen, M. H., Morris, B., Lieftink, C., Beijersbergen, R. L., van Leeuwen, E. M. M., & Kuijpers, T. W. (2020). High-throughput compound screen reveals mTOR inhibitors as potential therapeutics to reduce (auto)antibody production by human plasma cells. European journal of immunology, 50(1), 73–85. https://doi.org/10.1002/eji.201948241

Comment 2: Paragraph “Igs detection”. This method raises doubts. Firstly, after the stage of antigen sorption in the wells of the plates, there is no blocking stage. This significantly affects the result obtained. Second, anti-human polyvalent Ig (Cat. No. A3313) can react with adsorbed antibodies to human IgG (Cat. No. I1886) and/or antibodies to human IgM (Cat. No. I0104).
Response 2: Thank you for raising this important technical point. In classical ELISA protocols, a blocking step is often included after the antigen or antibody coating to prevent nonspecific binding to the plastic surface. However, in our assay, high-binding plates were used, and the wells were coated with highly specific monoclonal antibodies against human IgG or IgM, which effectively saturate the surface and act functionally as a blocking layer. This reduces the need for a separate blocking step. Regarding the detection antibody, the anti-human polyvalent Ig (Cat. No. A3313) can indeed bind to both IgG and IgM. However, because each well is specifically coated with either anti-IgG or anti-IgM capture antibodies, the detection antibody binds only to the respective immunoglobulin isotype captured in that well. This configuration minimizes cross-reactivity and nonspecific binding.
While this protocol has not been formally validated, it is widely used and accepted in immunotoxicology research, including results accepted by regulatory agencies such as the EFSA (Corsini et al., 2024). The assay setup and reagents used have been consistently applied in different published studies assessing total immunoglobulin secretion by human PBMCs in vitro (Corsini et al., 2024; Iulini et al., 2025).

Reference:
Corsini E, Iulini M, Galbiati V, Maddalon A, Pappalardo F, Russo G, Hoogenboom R, Beekmann K, Janssen A, Louisse J, Fragki S and Paini A, 2024. EFSA Project on the use of NAMs to explore the immunotoxicity of PFAS. EFSA supporting publication 2024: 21(8):EN-8926 146 pp. doi:10.2903/sp.efsa.2024.EN-8926
Iulini, M., Bettinsoli, V., Maddalon, A., Galbiati, V., Janssen, A. W. F., Beekmann, K., Russo, G., Pappalardo, F., Fragki, S., Paini, A., & Corsini, E. (2025). In vitro approaches to investigate the effect of chemicals on antibody production: the case study of PFASs. Archives of toxicology, 99(5), 2075–2086. https://doi.org/10.1007/s00204-025-03993-6

Comment 3: Line 177. Please specify which mathematical models were used. How were these mathematical models applied?
Response 3: Thank you for your request for clarification. The mathematical model used to estimate intracellular concentrations of BPA and its analogues from nominal in vitro concentrations is the IV-MBM EQP v2.0 model developed by Armitage et al. (2021). This high-throughput in vitro mass balance distribution model predicts the partitioning of chemicals between different compartments in cell culture systems, including intracellular space, based on physicochemical properties and experimental parameters. It allows for more accurate estimation of bioavailable intracellular doses rather than relying solely on nominal exposure concentrations. The application of this model enhances the relevance of in vitro findings to real biological exposure scenarios. We have now specified the exact mathematical model used in the study, namely the In Vitro Mass Balance Equilibrium Partitioning Model version 2.0 (IV-MBM EQP v2.0) developed by Armitage et al. (2021). This clarification has been added to the revised manuscript at line 211.
Reference: Armitage, J. M., Sangion, A., Parmar, R., Looky, A. B., & Arnot, J. A. (2021). Update and Evaluation of a High-Throughput In Vitro Mass Balance Distribution Model: IV-MBM EQP v2.0. Toxics, 9(11), 315. https://doi.org/10.3390/toxics9110315

Comment 4: Line 185. Please provide the reference and date of access to the PubChem resource
Response 4: Thank you. The reference to the PubChem database has been updated.

Comment 5: Line 199-207. “The content of storage lipids was 0.5%, while membrane lipids accounted for 2.5% of tissue volume. Structural proteins, specifically non-lipid organic matter, represented 0.10 of the cellular composition.” By what methods were these data obtained? What do they indicate and what is their significance in the context of this manuscript? These data do not characterize the mononuclear cells used in this study. A quantification of B cells that are involved in IgG secretion should be provided here.
Response 5: Thank you for this important observation. We agree that the values provided in this section refer to general cellular modeling parameters and do not specifically characterize our PBMC samples (or more specifically B cells). We have clarified this point in the manuscript to avoid any confusion (lines 234-236). Currently, there are no specific compositional data in the literature describing PBMCs in terms of detailed cellular fractions such as storage lipids, membrane lipids, or structural proteins. The values used in our modeling approach represent general cellular parameters commonly applied in this type of in vitro modeling. In particular, these same values were adopted in Corsini et al. (2024), where the IV-MBM EQP model was applied under similar experimental conditions, although with different test compounds. These values were selected by a panel of experts and are considered appropriate for simulating distribution dynamics in suspension cell systems such as PBMCs.
Unfortunately, we are not currently equipped to perform B-cell quantification within our experimental setup. Moreover, such a characterization would require dedicated experiments beyond the scope and timeframe of this revision, particularly considering that our exposure protocol spans 7 days. While we acknowledge the interest and value of including such data, we are unable to incorporate it at this stage.

Comment 6: Line 210-213. Please provide the method for propidium iodide (PI) staining and flow cytometry analysis in the Methods section. The flow cytometric results should also be provided and described in this manuscript. How was cell viability tested? Please also describe in the Methods section.
Response 6: Thank you for your comment. We have now added a detailed description of the propidium iodide (PI) staining and flow cytometry protocol used to assess cell viability in the Materials and Methods section (lines 161-168). Specifically, we included the procedure for PI staining, centrifugation, sample preparation, and analysis using the NovoCyte 3000 flow cytometer.
Additionally, the flow cytometry results used to determine non-cytotoxic concentration ranges (CV80) are now provided and described in the Supplementary Information (new Supplementary Figure S1). These data support the selection of concentration ranges used in the antibody secretion assays.

Comment 7: Line 237. What is the “stimulation index”? Please provide the antibody concentrations obtained in common units, such as μg/ml. The concentration of IgG produced by untreated cells should also be provided. Are there statistically significant differences between treated and untreated cells?
Response 7: Thank you for your insightful question. The Stimulation Index (SI) is defined as the ratio between the response measured in the treated (stimulated) condition and the corresponding response in the untreated control from the same donor. This approach is commonly used in immunological studies, especially when working with primary human cells, to account for individual variability. In our experiments, PBMCs were obtained from different donors, which naturally results in substantial variability in baseline antibody (IgG, IgM) and cytokine levels, typically expressed in ng/mL. These differences arise from factors such as genetic background, environmental exposures, and immune status. Because the absolute concentrations of antibodies produced by untreated cells varied widely among donors, reporting raw values alone could mask treatment effects due to high inter-individual variability. To address this, we normalized each donor’s treated values to their own untreated baseline, calculating the SI to highlight the relative change induced by the bisphenol exposure. This normalization enhances the comparability of results across donors and reduces variability that might obscure significant differences. We prefer to maintain the graphical representation using the SI, as it more clearly illustrates the relative effects across multiple donors. However, to provide additional clarity, we report the absolute antibody concentrations in ng/mL for non-stimulated (naive) cells compared to cells treated with vehicle control alone. This allows the reader to appreciate the baseline antibody production and the magnitude of stimulation in terms of actual IgG and IgM levels. This quantitative information are also integrated into the text at the beginning of the Results section to improve interpretability (lines 270-283).

Comment 8: Line 286. How does lactate dehydrogenase demonstrate cell viability? Please explain. This should also be described in the Methods section
Response 8: We agree this point requires clarification. The LDH release assay is commonly used to assess membrane integrity as an indirect measure of cytotoxicity. Its use is approved by OECD guidelines (Test Guideline 442D) for in vitro testing. In our context, low LDH release indicates preserved membrane integrity and supports the conclusion that immunoglobulin reduction was not due to overt cytotoxicity. A clarification has been added to the Methods section (lines 177-185).

Comment 9: Line 307. What is the mathematical model used based on? How can the differences in cell penetration for different compounds be explained? How is this related to their structure?
Response 9: Thank you for the question. The model used is based on physicochemical parameters (e.g., logP, molecular weight, topological surface area) obtained from PubChem, and assumes passive diffusion as the primary mode of cellular entry. Differences in intracellular concentrations can be partially explained by variations in lipophilicity, size, and polarity of each compound. These details have now been clarified in the revised manuscript at lines 338–350.

Comment 10: Line 322. Please clarify what plastic we are talking about?
Response 10: Thank you for the question. The term "plastic" refers specifically to the multiwell culture plates used during the in vitro exposure experiments. This is relevant because the in silico model accounts for chemical sorption to the plastic surface, which can influence the actual concentration of bisphenols available to cells. This clarification has been added to the revised manuscript at lines 357-366.

Reviewer 4 Report

Comments and Suggestions for Authors

This study is timely and interesting. 

The abstract clearly outlines the rationale, objectives, methodology, and main findings. It effectively emphasizes the concern that BPA analogues may not be safer alternatives. Strengths include a well-structured summary and clear mention of sex-specific differences, which adds novelty. However, it lacks detailed context regarding the concentrations tested and statistical significance, which could help assess biological relevance. Including specific ICâ‚…â‚€ and ICâ‚‚â‚€ ranges would improve interpretability. Additionally, the mechanisms underlying immunosuppression are not discussed, leaving readers with limited mechanistic insight. Overall, the abstract is informative and highlights the importance of evaluating BPA analogues for immunotoxicity, but could be strengthened by providing key numeric data and a brief mention of potential implications for human health.

I have just a few comments about Introduction and Discussion, other than adding some relevant new informations/references. 

I report them in bullet points:
- The introduction provides a comprehensive and up-to-date background on BPA as an endocrine disruptor and its immunomodulatory effects. It effectively integrates regulatory context (EFSA TDI revision) and highlights immune endpoints as critical targets, which strengthens the rationale for the study. The discussion of exposure routes, metabolism, and health outcomes is well-structured, though some sentences are long and could be simplified for clarity. Including both human and animal evidence for immunoglobulin modulation adds relevance, but references to Ig effects are somewhat fragmented; a summarizing sentence could improve flow.

-The gap regarding in vitro human studies on antibody production is clearly identified and aligns with the study objective. However, some numeric details (e.g., doses in μg/kg bw, μM) may overwhelm readers without context—consider emphasizing real-life exposure ranges earlier. Overall, the introduction is scientifically sound, well-referenced, and justifies the study, though minor stylistic refinements and better integration of analogues’ risk discussion would enhance readability.
- Accordingly, i suggest other relevant aspects: have you data about mouse model (e.g., EAE, PLP1 or Autoimmune encephalitis models?). I think it is relevant to add informations about models of autoantibodies mediated disorders in order to obtain more clinical data. 
- You should otherwise comment or create paragraphs about 1) Immune dysregulation impacting for example the nervous system (e.g., peripheral antibodies with neuro-pathogenic potential); 2) Sex-specific immune responses; 3) Role of Inflammatory cytokine modulation (e.g., TNF‑α, IL‑6). I suggest to add some relevant informations about CNS because you should mention how is the role on  BBB and why antibody-production reveal a sense of protection or disruption of blood brain barrier. This is a fundamental step in my opinion. 
- I didn't understand from which patients PBMCs were derived (only healthy controls ?). 
- Consider what kind of effect, from a clincial point of view, you can have with exposure. Memory impairment, Isolated memory deficits (please see  DOI: 10.1111/ene.70113 (Malvaso et al.) for a review about antibodies, clinico-demographic role and specific memory impairment related to autoantibodies, cognitive decline ? etc..). You should amplify this aspect from a neurological-autoimmunity point of view. 
- Moreover, you should create a parallelism with literature on which exposure to other agents could create an antibody-chronic-mediated syndrome (please see 10.1212/NXI.0000000000200314 and 10.3390/brainsci14080764 for an example and add some relevant literature). 
- Have you relevant data, from a clinical point of view, what is the impact of endocrine disruptors (EDs) to the environmental health, personality dimensions of people, self related health and mental health. In my opinion you could create additional rows (just a few) to consider also this aspect... maybe in the future creating specific questionnaires to understand better all the aspect of the effect exposure. About this, consider to add some other relevant examples: 10.3389/fpsyg.2022.923316, 10.3390/healthcare11111645 etc.) 

- Have you data about microglial activity? Some recent advances research demontrated similar effects on these cells in other neurodegenerative conditions (please see 10.3390/cells12242824). 

Discussion is fine but should double check for typos and try to consider to re-write it in a less boring format, especially for not expert readers. 

Author Response

Comments and Suggestions for Authors: This study is timely and interesting.

The abstract clearly outlines the rationale, objectives, methodology, and main findings. It effectively emphasizes the concern that BPA analogues may not be safer alternatives. Strengths include a well-structured summary and clear mention of sex-specific differences, which adds novelty. However, it lacks detailed context regarding the concentrations tested and statistical significance, which could help assess biological relevance. Including specific ICâ‚…â‚€ and ICâ‚‚â‚€ ranges would improve interpretability. Additionally, the mechanisms underlying immunosuppression are not discussed, leaving readers with limited mechanistic insight. Overall, the abstract is informative and highlights the importance of evaluating BPA analogues for immunotoxicity, but could be strengthened by providing key numeric data and a brief mention of potential implications for human health.

We appreciate your positive feedback on the structure and clarity of the abstract. We agree that including specific numeric values, such as the ICâ‚…â‚€ and ICâ‚‚â‚€ ranges, alongside statistical significance and mechanistic considerations, would enhance the interpretability of our findings and provide context for their biological relevance. However, due to the strict word/character limit for abstracts, it was not possible to include these additional details without compromising the summary's overall coherence and completeness. These aspects are thoroughly discussed and addressed in the main body of the manuscript, where both the numerical data and the mechanistic considerations are presented in full. We believe this approach ensures the abstract remains concise yet informative while directing readers to the relevant sections for further information.

I have just a few comments about Introduction and Discussion, other than adding some relevant new informations/references.

I report them in bullet points:

Comment 1: The introduction provides a comprehensive and up-to-date background on BPA as an endocrine disruptor and its immunomodulatory effects. It effectively integrates regulatory context (EFSA TDI revision) and highlights immune endpoints as critical targets, which strengthens the rationale for the study. The discussion of exposure routes, metabolism, and health outcomes is well-structured, though some sentences are long and could be simplified for clarity. Including both human and animal evidence for immunoglobulin modulation adds relevance, but references to Ig effects are somewhat fragmented; a summarizing sentence could improve flow.
Response 1: We thank the reviewer for the constructive and insightful comments. As suggested, we have revised the Introduction to improve clarity and readability by simplifying or splitting several long sentences. In addition, we have added a summarizing sentence to better integrate and contextualize the findings related to immunoglobulin (Ig) modulation by BPA and its analogues, in both human and animal studies (lines 79-85).

Comment 2: The gap regarding in vitro human studies on antibody production is clearly identified and aligns with the study objective. However, some numeric details (e.g., doses in μg/kg bw, μM) may overwhelm readers without context—consider emphasizing real-life exposure ranges earlier. Overall, the introduction is scientifically sound, well-referenced, and justifies the study, though minor stylistic refinements and better integration of analogues’ risk discussion would enhance readability.
Response 2: We appreciate the reviewer’s observations. Following the suggestion, we have clarified the relevance of the tested concentrations by explicitly including reported human exposure levels for BPA and its analogues in both plasma and urine (lines 45-51 and lines 113-117). This contextualization aims to better frame the in vitro dose selection and improve the accessibility of the information for readers. Additionally, we made minor stylistic refinements throughout the introduction to enhance clarity and flow, and we improved the integration of the discussion regarding the potential risks posed by BPA analogues.

Comment 3: Accordingly, i suggest other relevant aspects: have you data about mouse model (e.g., EAE, PLP1 or Autoimmune encephalitis models?). I think it is relevant to add informations about models of autoantibodies mediated disorders in order to obtain more clinical data.
Response 3: We thank the reviewer for this insightful suggestion and fully agree that in vivo models of autoimmune diseases—such as EAE, PLP1, or autoimmune encephalitis—could provide valuable mechanistic insight into the immunomodulatory effects of bisphenols, particularly in the context of autoantibody-mediated disorders. These models indeed represent a relevant direction for future research to better understand clinical implications. However, we are currently not in a position to include such data, as our research does not involve animal experimentation and focuses specifically on human in vitro systems. Therefore, while we acknowledge the importance of this approach, it falls outside the scope of the present study.

Comment 4: You should otherwise comment or create paragraphs about 1) Immune dysregulation impacting for example the nervous system (e.g., peripheral antibodies with neuro-pathogenic potential); 2) Sex-specific immune responses; 3) Role of Inflammatory cytokine modulation (e.g., TNF‑α, IL‑6). I suggest to add some relevant informations about CNS because you should mention how is the role on  BBB and why antibody-production reveal a sense of protection or disruption of blood brain barrier. This is a fundamental step in my opinion.
Response 4: We sincerely thank the reviewer for their thoughtful and detailed comments. We fully agree that immune dysregulation—particularly in relation to the central nervous system (CNS), peripheral antibody effects on the blood-brain barrier (BBB), sex-specific immune responses, and inflammatory cytokine modulation (e.g., TNF-α, IL-6)—represents a critical area of investigation. These aspects are highly relevant in the broader context of neuroimmunology and endocrine disruption. However, the scope of our current study was limited to an in vitro evaluation of immunoglobulin secretion by human PBMCs in response to bisphenol analogues. Our experimental design did not include CNS-related endpoints, such as BBB integrity, microglial function, or cytokine profiling, nor was it designed to assess systemic or neurological outcomes. Although we briefly mention some literature data regarding cytokine modulation and sex-based differences, we feel that a deeper discussion on CNS implications and BBB mechanisms would require dedicated in vivo models and a mechanistic framework that are beyond the aim and scope of this manuscript. We appreciate the reviewer’s suggestion and consider it a valuable direction for future multidisciplinary research integrating immunotoxicology with neurobiology.

Comment 5: I didn't understand from which patients PBMCs were derived (only healthy controls ?).
Response 5: We clarified in the Materials and Methods (line 153) section that PBMCs were isolated from five healthy male and female donors to specify the origin of the cells used.

Comment 6: Consider what kind of effect, from a clincial point of view, you can have with exposure. Memory impairment, Isolated memory deficits (please see  DOI: 10.1111/ene.70113 (Malvaso et al.) for a review about antibodies, clinico-demographic role and specific memory impairment related to autoantibodies, cognitive decline ? etc..). You should amplify this aspect from a neurological-autoimmunity point of view.
Response 6: We appreciate the reviewer’s insightful suggestion regarding the clinical implications of exposure to bisphenols, particularly memory impairment and antibody-related cognitive decline as discussed by Malvaso et al. While we recognize the importance of neurological-autoimmunity in the context of endocrine disruptors, our current study focuses on in vitro immunotoxicity endpoints related to antibody secretion by peripheral blood cells. Consequently, clinical outcomes such as memory deficits or neurocognitive impairments are beyond the scope of our experimental design and data. We acknowledge this limitation and suggest that further in vivo and clinical studies are required to elucidate these complex neurological-autoimmune connections

Comment 7: Moreover, you should create a parallelism with literature on which exposure to other agents could create an antibody-chronic-mediated syndrome (please see 10.1212/NXI.0000000000200314 and 10.3390/brainsci14080764 for an example and add some relevant literature).  
Response 7: We thank the reviewer for highlighting relevant literature on chronic antibody-mediated syndromes induced by various environmental agents. While drawing such parallels could enrich the discussion, the primary focus of our manuscript is on the immunotoxic potential of BPA and its analogues in vitro. Including an extensive comparative review on other agents would risk diluting the manuscript’s focus. However, we have briefly referenced related immunotoxic mechanisms and agree that future work could explore these parallels in greater depth.

Comment 8: Have you relevant data, from a clinical point of view, what is the impact of endocrine disruptors (EDs) to the environmental health, personality dimensions of people, self related health and mental health. In my opinion you could create additional rows (just a few) to consider also this aspect... maybe in the future creating specific questionnaires to understand better all the aspect of the effect exposure. About this, consider to add some other relevant examples: 10.3389/fpsyg.2022.923316, 10.3390/healthcare11111645 etc.)
Response 8: The suggestion to consider broader clinical aspects, including personality dimensions, self-related health, and mental health effects related to endocrine disruptor exposure, is highly valuable. Unfortunately, we do not possess clinical or epidemiological data addressing these complex endpoints within our study framework. Our research was confined to cellular immunotoxicology assays, and thus we did not include these dimensions. We concur that future multidisciplinary studies, potentially integrating validated questionnaires as suggested, could meaningfully contribute to understanding these impacts.

Comment 9: Have you data about microglial activity? Some recent advances research demontrated similar effects on these cells in other neurodegenerative conditions (please see 10.3390/cells12242824).
Response 9: We acknowledge the importance of microglial activation and its relevance to neurodegenerative diseases as highlighted by recent research. However, microglial activity was not evaluated in our current work since our model system utilized PBMCs, not CNS-resident cells. This aspect lies outside the scope of our in vitro approach but represents a promising avenue for future investigations addressing neuroimmune interactions of endocrine disruptors.

Discussion is fine but should double check for typos and try to consider to re-write it in a less boring format, especially for not expert readers. We carefully revised the Discussion section to reduce complexity and improve readability for a broader audience. Typos and minor errors were corrected throughout the manuscript.

Round 2

Reviewer 3 Report

Comments and Suggestions for Authors

The new additions to the manuscript made a big difference. The quality of the paper had improved, and all my questions were addressed. No more comments.